# Landscape Aesthetic Value of Waterfront Green Space Based on Space–Psychology–Behavior Dimension: A Case Study along Qiantang River (Hangzhou Section)

**DOI:** 10.3390/ijerph20043115

**Published:** 2023-02-10

**Authors:** Xiaojia Liu, Xi Chen, Yan Huang, Weihong Wang, Mingkan Zhang, Yang Jin

**Affiliations:** 1School of Design and Architecture, Zhejiang University of Technology, Hangzhou 310023, China; 2School of Landscape Architecture, Zhejiang Agriculture and Forestry University, Hangzhou 311302, China; 3Institute of Computer Network Application, Zhejiang University of Technology, Hangzhou 310023, China

**Keywords:** urban waterfront spaces, landscape aesthetic value, spatial–psychological–behavioral dimension, horizontal–vertical–three-dimensional space evaluation model, heat map, behavior observation

## Abstract

As an important part of urban green infrastructure, the landscape effect of the urban waterfront green space varies, and sometimes, the green space with an excellent landscape aesthetic value fails to serve the needs of most citizens. This seriously affects the construction of a green ecological civilization and the implementation of the concept of “common prosperity” in China. Based on multi-source data, this study took the Qiantang River Basin as an example, selected 12 representative waterfront green spaces along the river as the research objects, and used qualitative and quantitative analysis methods to determine the landscape aesthetic value of the research area from the different dimensions of space, psychology, and physiology. We examined the relationship between each dimension so as to objectively and comprehensively reflect the landscape value characteristics of the waterfront green space in the study area and provide a reasonable theoretical framework and practical development path for future urban waterfront green space landscape design. We obtained the following results: (1) The results of the spatial dimension research indicated that the spatial value index of the waterfront green space in the study area was three-dimensional space > vertical space > horizontal space, and the overall spatial value was low; Qianjiang Ecological Park obtained the highest value (0.5473), and Urban Balcony Park obtained the lowest value (0.4619). (2) The results of the psychological dimension indicated that people’s perceptions of the waterfront green space in the study area were relatively weak, mainly focusing on visual perception, but the waterfront green space with a relative emotional value greater than one accounted for 75%, and the overall recognition of the landscape was high. (3) The results of the behavioral dimension showed that the overall heat of the waterfront green space in the study area was insufficient (1.3719–7.1583), which was mainly concentrated in low-heat levels, and the population density was unevenly distributed (0.0014–0.0663), which was mainly concentrated in the medium-density level. The main purpose of users was to visit, and they stayed an average of 1.5 h. (4) The results of the coupling coordination analysis of the spatial–psychological–behavioral dimensions showed that the landscape value of the waterfront green space in the study area presented a form of ‘high coupling degree and low coordination degree’.

## 1. Introduction

As an important part of urban green infrastructure, waterfront green space has multiple values, including a natural society. From the perspective of natural value, urban waterfront green space is located in the land–water transition zone, and as an ecological buffer zone, it has ecological functions such as soil and water conservation, microclimate regulation, air purification, enhancement, and protection of biodiversity [1,2,3]. From the perspective of social value, urban waterfront green space, as a window of modern cities and an important symbol of civilization, can significantly improve the urban landscape, drive the surrounding economic development, optimize the living environment, and have many positive impacts on human health and well-being [4,5]. For example, coastal ecosystems tend to be characterized by high human well-being, clean water, clean air, and aesthetic pleasure, and the jogging and walking space on the waterfront gives it the durability, stability, and productivity of a natural system [6]. From the perspective of spatial division, waterfront projects use the blue line (water protection zone control line) and the green line (urban green space control line) as the main control lines to divide ‘blue space’ and ‘green space’ [7,8,9]. The ‘waterfront green space’ represents a mixed location of ecological, economic, and social transition and diffusion areas and is spatially concretized between land and water [1]. However, at this stage, sometimes the landscape effect of urban waterfront green space varies, and the green space with excellent landscape aesthetic value fails to serve the needs of most citizens. This leads to the uneven distribution of public social resources [10,11,12,13,14], the simple pursuit of modernization and standardization of waterfront landscape [15,16], and the intensification of contradictions between different types of land use in waterfront areas [17,18,19,20,21,22]. These phenomena are not conducive to China’s development of green ecological civilization construction ideas or China’s common prosperity concept. Therefore, it is necessary to explore the multi-landscape of both green space and blue space and optimize the organization of the waterfront landscape to enhance the aesthetic value of urban waterfront green space [23,24].

The study of landscape aesthetics began in the late 1960s [25,26,27]. The research methods are divided into subjective paradigm (based on perception and public preference) and objective paradigm (based on landscape metrics to evaluate the objective visual value quality of landscape) [28,29,30,31,32,33,34,35]. With the development of landscape aesthetics research, nowadays, landscape aesthetics evaluation mainly relies on spatial, psychological, and physiological perspectives. Among them, landscape space aesthetics evaluation methods are mainly based on the quantitative analysis of sites based on remote sensing image data [36,37,38,39,40,41], such as environmental simulation prediction models to quantify the aesthetic quality of urban waterfront space [42] and mobile LiDAR point clouds to draw three-dimensional models of urban street environments [43]. Psychological and aesthetic evaluation is mainly based on crowd perception [44,45], using public participation, semantic analysis, and other methods to obtain the results of the crowd’s emotional perceptions to determine the value quality of landscape aesthetics [46,47]. For example, social media texts are used to explore the emotional reactions of users to waterfront landscapes [48], and urban landscape perception research is based on public streetscape images [49]. Behavioral aesthetics evaluation is mainly conducted through field research and other methods to obtain multi-source big data, to study the crowd recreation behavior, and thus to reflect the aesthetic value of the study area [50]. For example, studies of user behavior characteristics were based on behavior observation and GIS [51,52] and analyzed users’ responses to the Washington Cherry Blossom Festival through their behavioral motivation [53]. However, because of the comprehensive influence of natural and social factors such as the regional economy and surrounding environment, the existing landscape aesthetics research cannot unify the dimensions of space, behavior, and psychology [27,33]. Only a single-level analysis can fully and accurately evaluate the aesthetic characteristics of the site, reflecting its aesthetic value; therefore, we must build a more scientific, systematic, and humane multi-dimensional landscape aesthetic evaluation system [54,55,56].

The Qiantang River is the largest river in Zhejiang Province, China. It is the birthplace of Wuyue culture. It has a long history and important cultural, ecological, and economic values. There are a variety of urban waterfront green space landscapes on both sides of the river, attracting many people for leisure and relaxation. Therefore, this study took the Hangzhou section of the Qiantang River as the research scope and selected 12 waterfront parks on both sides of the river as the specific research objects. Aiming to investigate the problems of unreasonable spatial element structure, the unmet unsatisfactory viewing needs of leisure travelers, and their low recognition of waterfront green space, we constructed a theoretical framework of landscape aesthetics based on space–psychology–behavior and comprehensively summarized the aesthetic characteristics of the waterfront green space landscape in the study area. This will provide a reasonable and feasible theoretical framework and practical development path for the Qiantang River waterfront green landscape design.

## 2. Materials and Methods

### 2.1. Study Area

With advances in urbanization and industrial upgrading and transformation in Hangzhou (the capital city of Zhejiang Province, China), the urban center has gradually shifted eastward. This has been developed as the central activity area planned by Hangzhou and the main center of the city. Hangzhou has shifted from the ‘West Lake era’ to the ‘Qiantang River era’. As an important urban ecological corridor throughout the whole city of Hangzhou, the Qiantang River (the downstream waterway of Wenjiayan in Hangzhou) is the leading linear landscape element of urban development. In this study, the urban rapid ring road around the river was designated as the research area (Figure 1), and 12 representative waterfront green spaces along the coast were selected as the research objects (Table 1), namely, Qianjiang Century Park (QCP), Qianjiang Ecological Park (QEC), Yanjiang Wetland Park (YWP), National Water Museum of China (NWMC), Yanjiang Park (YP), CBD Park (CP), Wangjiang Park (WP), Baita Park (BTP), Binjiang Park (BP), Liuheta Cultural Park (LCP), Wenyan Riverside Park (WRP), and Urban Balcony Park (UBP). The selected objects are all open parks (Figure 2). According to the division of their administrative regions, they span five administrative regions in Hangzhou: Xiaoshan District (NWMC, QCP, YP, WRP), Qiantang District (YWP, QEP), Binjiang District (BP), Shangcheng District (CP, BTP, WP, UBP), and West Lake District (LCP). The Ministry of Housing and Urban-Rural Development of China in 2017 issued the urban green space classification standard in the park green space category: there are 7 comprehensive parks, including NWMC, QCP, CP, QEP, BTP, UBP, WRP; 2 special parks, including LCP (Heritage Park) and YWP (Urban Wetland Park); 2 linear parks, including BP and YP; 1 green space beside the street, which is WP. According to the location relationship between parks and the water, 7 parks are located on the north bank of the river, and 5 parks are located on the south bank of the river.

### 2.2. Landscape Aesthetic Evaluation Method

This study was based on multi-source data and combined quantitative and qualitative research methods to study the aesthetic value of the research area regarding three dimensions: space, behavior, and psychology. On this basis, we made an in-depth study of the relationship between space, behavior, and psychology and analyzed its internal reasons (Figure 3). Figure 3 shows the index evaluation results of the spatial dimension, which were used as the preliminary information for the aesthetic characteristics of the waterfront park. This system integrated the spatial–psychological–behavioral dimension to evaluate the aesthetic characteristics of the waterfront parks.

#### 2.2.1. Spatial Dimension of Landscape Aesthetics Research

This study established a spatial aesthetic evaluation system based on horizontal, vertical, and three-dimensional spaces (Table 2) [57,58]. The horizontal space contained the four indicators of species diversity, landscape evenness index, landscape disturbance index, and landscape fragmentation index; the vertical space included four indexes: vertical visual level, color diversity, slope shape, and site section geometry. Three-dimensional space included four indicators: diffuse non-interceptance (DIFN), tridimensional green biomass, canopy density, and relief amplitude. According to the normalization of each index, the geometric average was taken to obtain the different spaces and overall aesthetic value of the green landscape space in the study area.

#### 2.2.2. Psychological Dimension of Landscape Aesthetics Research

Based on text sentiment analysis technology [71,72,73], this study evaluated the landscape aesthetics of the research object from the psychological point of view of users. The specific steps were as follows.

First, by calculating the Weibo evaluation related to the landscape experience, we derived the landscape emotional value and park emotional value. After conversion by text emotion technology, the value was between 0 and 10. When the value was greater than 5, the experience had a positive emotional perception. The greater the value, the higher the positive degree of experience perception.

Second, the relative emotional value was obtained from the ratio of the landscape emotional value to the park emotional value so as to exclude the influence of the park itself on the users and obtain the attractiveness evaluation of the park landscape. When the relative emotional value was greater than 1, the landscape had a positive attraction perception. The larger the value, the greater the attraction of the park landscape to visitors and the higher the recognition of it.

Subsequently, this study used linear regression analysis to verify the relationship between visitors’ landscape experiences and emotional perceptions [74]. The landscape relative evaluation rate represented the landscape experience as an independent variable, while the relative emotional value represented the emotional perception as a dependent variable. If the significance (p) of the landscape relative evaluation rate was less than 0.05, the visitors’ landscape experiences and emotional perceptions had a positive impact relationship. The closer the R^2^ of the fitting situation was to 1, the stronger the linear correlation between the two variables.

Finally, in order to further analyze the characteristics and differences in users’ sensory evaluations, this study conducted word frequency analysis on the parks with emotional values greater than 1 based on the results of relative emotional values. The word frequency analysis technology was used to segment, check, and fit the evaluation of the park [75,76], and people’s emotional perceptions were obtained from the five aspects of vision, smell, hearing, touch, and taste, so as to analyze the results of the crowd experience of different waterfront spaces.

#### 2.2.3. Behavioral Dimension of Landscape Aesthetics Research

From the perspective of user behavior, this study was based on a Baidu heat map [77,78], nuclear density method [79,80,81], recreational behavior observation method [82], and the social attributes and recreational behaviors of different groups in the study area, thus reflecting the landscape aesthetic value of the waterfront park. The specific steps were as follows.

First, the thermal population distribution of waterfront park was obtained by calculating the thermal value with ArcGIS 10.7.

Second, according to the results of behavior observation, the recreation time, number, frequency, and purpose of park visitors were obtained.

Finally, using the above results, the study calculated and drew the nuclear density map of the park population in ArcGIS 10.7 and obtained the nuclear density values of different parks, so as to analyze the behavior and environmental preferences of users.

### 2.3. Data Collection and Preprocessing

Land-use data: This study purchased the latest land-use data of Hangzhou in 2020 from the Esri official website, with an accuracy of 10 M.

Elevation data: According to ALOS satellite (Advanced Land Observation Satellite), the Hangzhou Digital Elevation Model (DEM) was obtained with an accuracy of 12.5 M.

Panoramic photos: This study obtained panoramic photos by the grid method in the park space. Because of the large area difference between the research objects, we divided the park into two categories according to the area. The park area within 290,000 square meters was divided into 15 m × 15 m equal grid and the park area larger than 290,000 square meters was a 30 m × 30 m equal grid. We shot in the center of each grid. We took the direction of the park route as a perspective for the panoramic photos, which was consistent with the residual light perspective of the human eye (horizontal 180 degrees, vertical 90 degrees). The shooting tool selected the panoramic photography mode of the Apple mobile phone, and we used the tripod to keep the shooting stable. Shooting height refers to the average height of the line of sight, which was set at 165 cm (Figure 4). The shooting dates were 24 August to 12 September 2022, and the shooting time was unified between 9:30–11:30 in the morning and 1:30–4:30 in the afternoon when the weather was sunny. According to the shooting points investigated, we finally selected a total of 625 effective photos.

Spatial dimension data preprocessing: We first extracted the land use data and elevation data of each park in ArcGIS 10.7, put the land use data of the park into the calculation results of Fragstats, entered them into ArcGIS 10.7 to obtain C1, C2, C3, and C4, and then used the elevation data to calculate C7, C8, and C12 in ArcGIS 10.7. Subsequently, C5, C6, C9, C10, and C11 were calculated in Photoshop 2022 with the park panorama and plan. Finally, all the results were normalized to obtain the final value. The evaluation results were divided into four grades by natural breakpoint method. The higher the grade, the higher the landscape space value of the park.

Weibo comments: This study used Octopus (a web information collection software) to obtain the relevant microblog evaluation of the waterfront park over the past five years for analysis. Specific content included user ID, gender, check-in point, release time, and evaluation text. Octopus software was used to crawl the data, and 6854 Weibo evaluations were obtained. Over 3 days, we eliminated redundant information, such as meaningless evaluations and advertisements, and finally obtained 4454 valid data.

Crowd thermal data: Real-time data were obtained according to the Baidu heat map. The dates were 12–18 August 2022. Nine heat maps were intercepted every day, one at 8:00, 9:00, and 10:00, one at 2:00, 3:00, and 4:00, and one at 19:00, 20:00, and 21:00.

Crowd recreation behavior data: This study used the user behavior observation method to obtain the crowd entertainment behavior data. The survey dates were from 20 July to 20 August 2022. The observation time was from 8:00 to 19:00. The observation method included the total number of visitors entering the park per hour at the main entrance. We asked users at the entrance to the park how much time they planned to spend visiting the park and how often they visited, and another researcher observed visitor behavior in the park.

## 3. Results

### 3.1. Spatial Dimension of Landscape Aesthetics Research

The Hangzhou section of the Qiantang River waterfront park space index results was mainly concentrated in the 0.4619–0.5473 range (Table 3). Among them, the landscape space value of QEP was the highest, and the UBP landscape space value was the lowest. According to the evaluation grade analysis, the aesthetic value of the waterfront park space was mainly concentrated in the second level, accounting for 41.6% (the number of second-level parks accounting for the proportion of all parks) and comprising YP, WRP, YWP, BTP, and NWMC; the third level included CP, BP, LCP, accounting for 25% (the number of third level parks accounting for the proportion of all parks); the first-level and fourth-level parks were the least distributed, accounting for 16.7% (the number of first-level and fourth-level parks of the same accounting for the proportion of all parks). The overall aesthetic value of waterfront open green space in the Hangzhou section of the Qiantang River was low, and the distribution of spatial aesthetic value was uneven.

In order to further analyze the heterogeneity of the regional aesthetic landscape, the characteristics of the aesthetic value of the horizontal–vertical–three-dimensional spatial landscape were analyzed (Table 4).

According to the mean value of the calculated index results, the horizontal space results were distributed between 0.3316 and 0.5944. Among them, more than 0.5 parks, 0.4–0.5 parks, and less than 0.4 parks had the same ratio as the total number of parks. The horizontal space result of UBP was the lowest of all results, with a value of 0.3316. The vertical spatial results were distributed between 0.4047 and 0.4921; UBP had the highest value and QEP the lowest. The three-dimensional spatial results were distributed between 0.4932 and 0.6987. Among them, the three-dimensional spatial value of CP was the highest, and the value of NWMC was the lowest. Overall, the parks’ horizontal space numerical differentiation was most obvious, followed by three-dimensional space; the vertical space differentiation was small. Among them, the aesthetic value of three-dimensional space was relatively good, and the aesthetic value of vertical space was poor.

### 3.2. Psychological Dimension of Landscape Aesthetics Research

According to the results of sentiment text analysis (Table 5), the landscape emotional values of QCP, YWP, and BTP were greater than 5, accounting for 25%. YWP had the highest landscape emotional value of 7.8. Park emotional values were distributed between 4.5 and 0.8.

According to the analysis of relative emotional value, the parks with a value greater than one were QCP, YWP, NWMC, YP, CP, BTP, BP, LCP, and WRP, accounting for 75%, indicating that these park landscapes had a positive attractiveness to visitors. Among them, the relative emotional values of QCP and BTP were greater than 2, which indicated that visitors had a strong perception of the two park landscapes. The park landscape had a strong attraction for visitors, and visitors had a high recognition of the park landscape.

According to the results of the linear fitting analysis, there was a significant linear positive correlation between the landscape relative evaluation rate and the relative emotional value (Figure 5), *p* = 0.028, R^2^ = 0.398. It can be seen that the stronger the experience of the park landscape, the better the emotional perception and the higher the recognition of the park landscape.

Parks with relative emotional values greater than two (Qianjiang Century Park, Baita Park) had more terms related to all senses and feelings because people commented on them the most (Table 6). There were some differences in the sensory experience of parks (such as Wenyan Riverside Park, Yanjiang Park, and Yanjiang Wetland Park) with emotional values between one and two. Visitors had a good feeling about the park landscape; most perceptions were positive, such as those reflecting comfort and happiness. Among them, visual perception was stronger, and the other four senses were less evaluated, indicating that the five senses of visitors in the waterfront park were not balanced.

### 3.3. Behavioral Dimension of Landscape Aesthetics Research

According to the average thermal value of the study area (Table 7). Using the natural breakpoint method, the areas with a heat intensity of 5.0001–8 are collectively referred to as a high-heat area, the areas with a heat intensity of 2.0001–5 are collectively referred to as a sub-heat area, and the areas of 1–2 are collectively referred to as a low-heat area. The results indicated that the average thermal value of the research object was concentrated from 1.3719 to 7.1583. According to the analysis of evaluation grade, the distribution of thermal grade was mainly concentrated in the low-heat grade, accounting for 50% of the total parks. The high-grade parks were CP, UBP, and QEP, accounting for 25%. Therefore, the waterfront park had insufficient heat and uneven distribution.

The results of user behavior observation (Table 8) indicated that a total of 23 kinds of activities were recorded over eight observation days. According to the characteristics and frequency of activities, the frequency of users’ recreational purposes from high to low was sightseeing, fitness, social networking, catering, and others. According to the analysis of recreation time, the average recreation time of users was 1–2 h, followed by 0.5 h. Among them, users who were motivated to visit (66.6%) mostly preferred to walk, take a sightseeing bus, or use a multi-bike, so they stayed longer, mostly 1.5 h or more.

In order to further study the distribution of users’ environmental preferences, this study conducted a nuclear density analysis based on the Baidu heat map and user behavior observation analysis. The results (Figure 6) indicated that the density of users was 0.0014–0.0663 persons/m^2^, which was divided into high, medium, and low density according to the natural breakpoint method. The kernel density of the waterfront green space in the study area was concentrated in the low-density level, ranging from 0.0014 to 0.0197, accounting for 41.7% of the total parks. The high-density green spaces were CP, QCP, and LCP, accounting for 25% of the total parks. The crowd concentration of UBP, WRP, QEP, and BTP was medium density level. Thus, the study area waterfront park population density was low.

## 4. Discussion

### 4.1. The Interaction among Space, Psychology and Behavior Dimensions

Urban riverside green space is not only an important part of the urban ecosystem but also an important carrier of urban civilization. It is also a green space for people to visit and relax in every day; at the same time, the ecosystem of waterfront green space provides a series of services, including economic benefits, improved air quality, and improved physical and social well-being and mental health [83]. Studies have shown that people’s psychology and behavior are closely related to spatial quality, and the spatial quality of an excellent landscape has a positive impact on people’s social lives, psychology, and behavior [84]. The research on the relationship between user psychology and behavior is useful for public space design [85].

We live in an extremely complex and interconnected world. People’s psychological perceptions can reflect, adapt, and change the world around them, and they have an important influence on their behavior [86]. At present, it can be proved that people’s psychology and behaviors are affected by their perceptions [87]. Therefore, we regard them as a perception system. If we assume that spatial quality can drive an individual’s perception system, we must consider that the quality of the entire space can be changed with changes in those landscape elements. At the same time, in order to pursue a higher aesthetic quality, we must further explore the coupling and coordination relationship between landscape space and people’s psychological and behavioral perceptions and analyze the degree of coupling between the different dimensions.

### 4.2. The Coupling Coordination Results of Spatial, Psychological and Behavioral Dimensions

First of all, according to the space–psychology–behavior dimension coupling coordination model analysis (Figure 7), the coupled coordination degree of this dimension of the parks was not balanced (Table 9), with a coordination level distribution at the 3–9 level, showing a ‘high degree of coupling, low coordination degree’ form. QCP was the park with the highest aesthetic value under the spatial–psychological–behavior dimension. The park had the spatial characteristics of high biodiversity, color diversity, and tridimensional green biomass. In the study of users’ psychological perceptions and behavior, it was proven that users had a positive perception of these spatial characteristics. At the same time, the study found that the calculation results of the space–psychology–behavior dimension of the park were relatively high and balanced. Studies have shown that people have a positive and healthy response to the space when they perceive the benefits of the ecosystem. Improving the space quality according to the characteristics of the indicators can improve people’s satisfaction preference for space so that users experience effective and positive psychology and behavior [88], which accounts for the good coupling and coordination of the park. WP was the park with the lowest comprehensive aesthetic value. The study found that the park’s low species diversity and canopy density had a negative impact on public sentiment, and there were phenomena such as neglect of ecological landscape construction and weak management intensity. At the same time, the calculation results of the spatial–psychological–behavior dimension of the park varied greatly, which explained the moderate imbalance of the coupling coordination degree. Considering the influence of spatial dimension factors on users’ moods, the construction department should strictly plan and manage the green space landscape, actively solve the problems that hinder the ecological development of green space, and bring a superior aesthetic experience to the public.

Second, the coupling relationship of the spatial dimension, psychological dimension, and behavior dimension of the waterfront park presented differentiated distributions in the results (Table 10, Table 11 and Table 12). The research indicated that (1) QCP, YP, WRP, YWP, and BTP had a strong spatial–psychological coupling. These parks had good overall landscape quality and high interaction with the landscape to enhance the positive emotions of users, and visual sensory factors such as good sky visibility and rich colors also improved the emotional perceptions of users, which was conducive to their physical and mental health, reduced depression, and made them feel happy. (2) QCP, CP, WRP, BTP, and LCP had a strong coupling degree of the psychological–behavioral dimensions. Behavioral observation studies have found that there is a positive correlation between landscape perception and recreation frequency, which explains the relatively high nuclear density of the park. (3) QCP, QEP, YWP, and CP had a strong coupling degree of the space–behavior dimension. Through analysis, we found that these parks had a good spatial quality to enrich users’ recreation behavior and diversify their recreation behavior.

### 4.3. Future Recommendations and Deficiency

This study analyzed the landscape aesthetic value of the study area from the spatial–psychological–behavioral dimension. Importantly, we determined the landscape aesthetic value of different waterfront parks so as to provide relevant guidance for stakeholders according to regional differences and actual conditions. For low aesthetic landscape parks, when improving the quality of space, we should further strengthen the supervision mechanism and establish a hierarchical risk management strategy [89] so as to minimize the negative effects caused by human beings and improve the wetland ecosystem services function of the city [90,91]. In addition, cities with wetlands should determine vacant areas of wetland protection based on the existing important wetland landscape [92].

This study has limitations. First of all, Sina Weibo users are mostly young people, which affected the statistical accuracy of waterfront park visitors’ emotions. Second, the calculation results of panoramic images do not fully represent the visual experience of pedestrians and may affect the overall rating. Therefore, more advanced technologies (artificial intelligence combined with big geographic data) will be used in future research to improve the aesthetic quality data of the waterfront landscape space. Some of the calculations are based on very few opinions about the parks (Table 5); this, too, should be verified in the future as the results obtained so far may not be very representative. In addition, we were affected by the epidemic when we investigated, and there was no great difference in recreation behavior between different groups of people, such as tourists and local residents. In the future, we will distinguish between park users with different attributes and further study the differences in user behavior patterns.

## 5. Conclusions

This study took the Qiantang River Basin as an example and, based on multi-source data, selected 12 representative waterfront green spaces along the coast as the research objects. We used comprehensive qualitative and quantitative analysis methods, analyzed the landscape aesthetic value of the research area from the different dimensions of space, psychology, and physiology, and investigated the relationship between each dimension so as to objectively and comprehensively reflect the landscape value characteristics of the waterfront green space in the research area. The spatial dimension landscape aesthetic evaluation, based on land-use data, elevation data, site panoramic images, and other data, established a horizontal–vertical–three-dimensional evaluation model to evaluate the spatial aesthetic value; the psychological dimension landscape aesthetic evaluation was based on Weibo evaluation information, using text sentiment analysis and word frequency analysis technology. The behavior dimension landscape aesthetic value was based on Baidu thermal data, the integrated use of behavior observation, and kernel density analysis. The main findings are as follows:

(1)The spatial dimension of landscape aesthetics research indicated that the overall aesthetic value of the waterfront green space in the study area was low. Among them, the QEP space aesthetic value index was the highest, and the lowest was UBP. According to the evaluation grade analysis, the study area waterfront park space aesthetic value was mainly concentrated in the secondary level. The spatial aesthetic value of the waterfront park was three-dimensional space > vertical space > horizontal space.(2)The psychological dimension of landscape aesthetics research indicated that people’s perceptions of the waterfront green space in the study area were relatively weak. The waterfront green space landscape had a positive attraction for visitors, and the waterfront green space with a relative emotional value greater than one accounted for 75%. The higher the relative emotional value of the waterfront green space landscape, the richer and more comprehensive the five-sense sensory experience of users, which also has a connection to a large number of comments available about the park. Among them, QCP had the highest relative emotional value, and WP had the lowest.(3)The behavioral dimension of landscape aesthetics research indicated that the overall thermal value of the waterfront green space in the study area was insufficient and, distributed from 1.3719 to 7.1583, was mainly concentrated in low-heat levels. Users were mostly motivated by sightseeing. The population density of the waterfront green space in the study area was generally low, and the kernel density value was distributed in the 0.0014–0.0663 range, mainly concentrated in the low-density level. CP had the highest population density, and NWMC had the lowest population density.(4)The results of the coupling coordination analysis indicated that the distribution of the coupling coordination degrees of the waterfront green space in the study area was not balanced, showing the form of ‘high coupling degree and low coordination degree’. Among them, the spatial–behavior–psychological coupling coordination degree of QCP was the best, and the spatial–behavior–psychological coupling coordination degree of WP was the lowest. The spatial–psychological dimension coupling degrees of QCP, YP, WRP, YWP, and BTP were strong. The coupling degrees of the psychological–behavior dimension of QCP, CP, WRP, BTP, and LCP were strong. The coupling degrees of the space–behavior dimension of QCP, QEP, YWP, and CP were strong.(5)This study avoided the shortcomings of fewer data points, high cost, and the low efficiency of traditional research methods, providing a practical value evaluation method for future landscape aesthetics.

## Figures and Tables

**Figure 1 ijerph-20-03115-f001:**
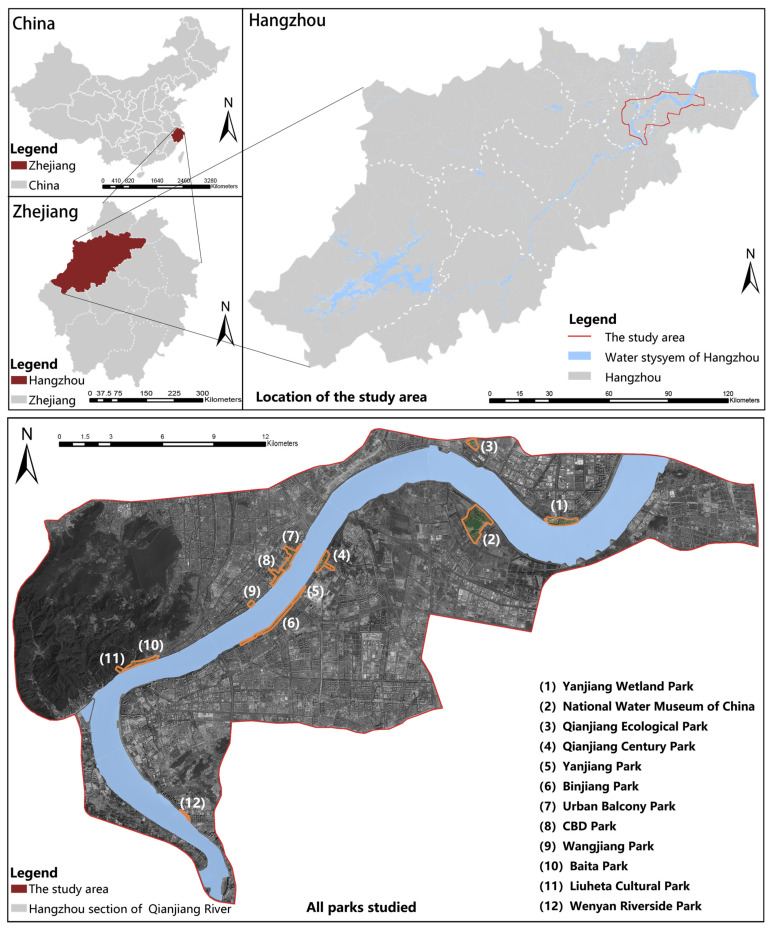
Location and distribution of waterfront parks.

**Figure 2 ijerph-20-03115-f002:**
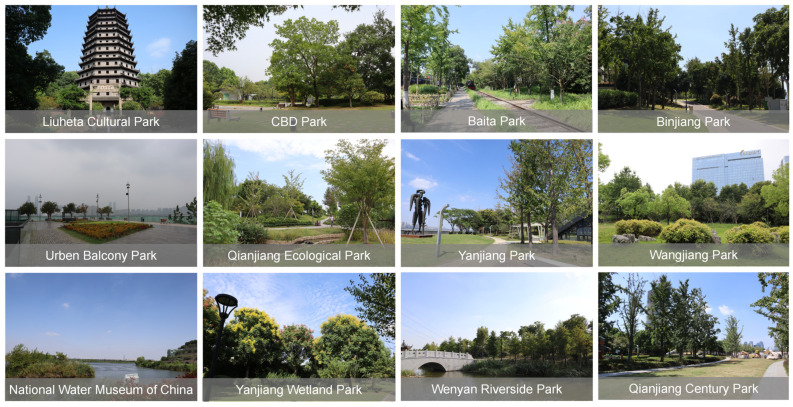
Present situation of waterfront parks.

**Figure 3 ijerph-20-03115-f003:**
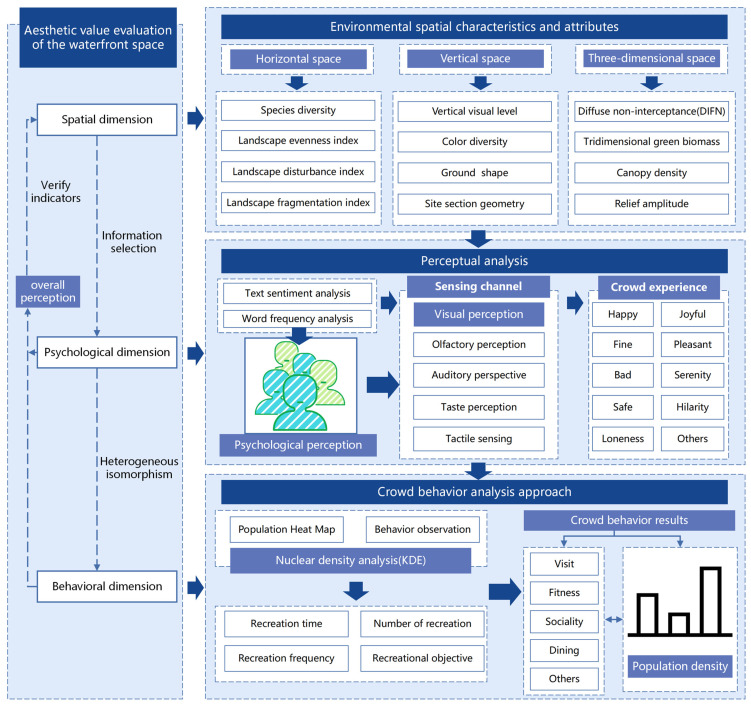
Space–psychology–behavior dimension landscape aesthetic evaluation mechanism.

**Figure 4 ijerph-20-03115-f004:**
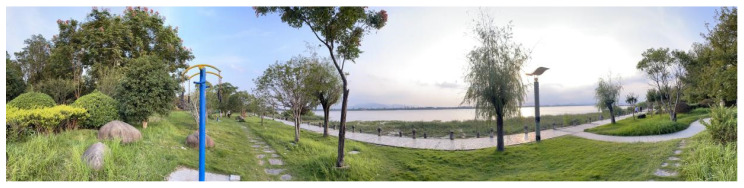
Panoramic picture sample.

**Figure 5 ijerph-20-03115-f005:**
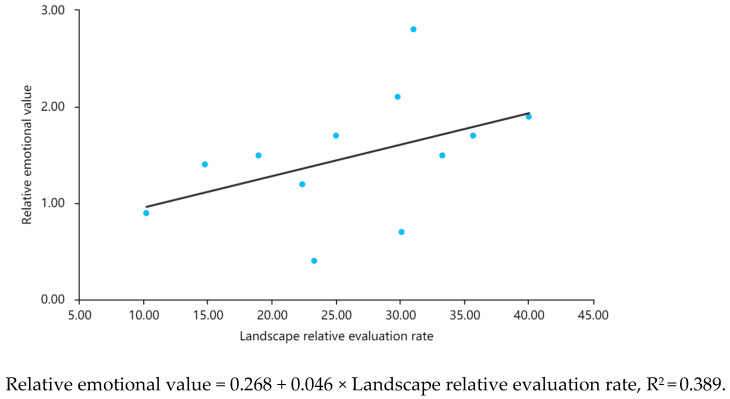
Scatter diagram of relative evaluation rates and relative emotion values of the landscape.

**Figure 6 ijerph-20-03115-f006:**
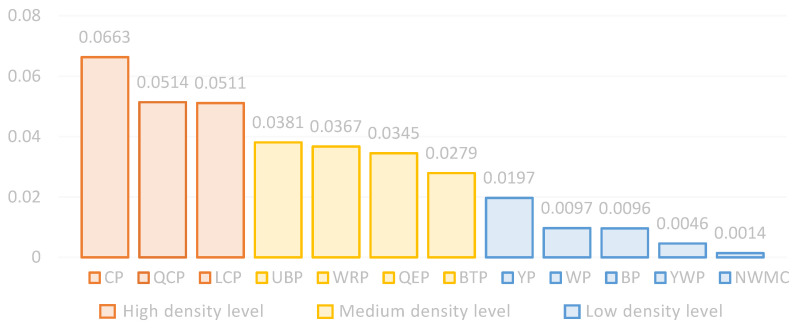
Park kernel density estimation mean data (persons/m^2^).

**Figure 7 ijerph-20-03115-f007:**
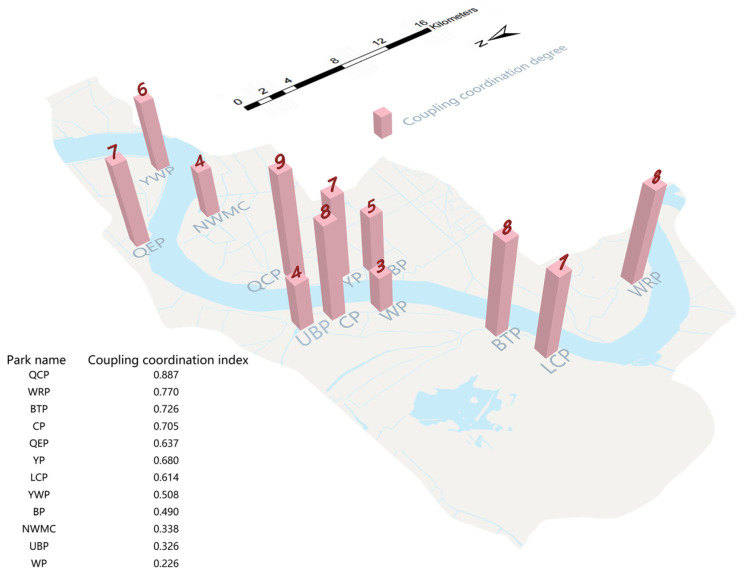
Coupling coordination degree distribution of waterfront parks.

**Table 1 ijerph-20-03115-t001:** Location and area of the waterfront parks (Hangzhou City, Zhejiang Province).

**Park Name**	**Location**	**Geographical Coordinates (N, E)**	**Area (HA)**
NATIONAL WATER MUSEUM OF CHINA (NWMC)	No. 1 Shuibo Avenue, Xiaoshan District	120.32033, 30.26778	129.8
QIANJIANG CENTURY PARK (QCP)	Guanlan Road, Xiaoshan District	112.45942, 34.62426	62
YANJIANG WETLAND PARK (YWP)	The intersection of No. 15 Avenue and Zhijiang East Road in Qiantang District	120.36557, 30.26874	33.2
BINJIANG PARK (BP)	Wentao Road Intersection, Jianghan Road, Binjiang District	120.20504, 30.21559	29.07
CBD PARK (CP)	No. 181 Wuxing Road, Shangcheng District	112.45942, 34.62426	26
YANJIANG PARK (YP)	Wentao Road, Xiaoshan District	120.23452, 30.24166	19.06
QIANJIANG ECOLOGICAL PARK (QEP)	No. 270 Qige Road, Qiantang District	120.32112, 30.30381	18
BAITA PARK (BP)	No. 4, Old Fuxing Street, Shangcheng District	120.14627, 30.20504	15.66
LIUHETA CULTURAL PARK (LCP)	No. 16 Zhijiang Road, Xihu District	120.14079, 30.20339	10.8
WENYAN RIVERSIDE PARK (WRP)	Intersection of Wanda North Road and Wanda Middle Road, Xiaoshan District	120.17122, 30.13673	3.98
WANGJIANG PARK (WP)	No. 844 Zhijiang Road, Shangcheng District	120.20483, 30.23133	3.32
URBAN BALCONY PARK (UBP)	No. 1078 Zhijiang Road, Shangcheng District	120.22326, 30.24792	2.81

**Table 2 ijerph-20-03115-t002:** Waterfront landscape space aesthetic value evaluation index.

Goal Layer (A)	Criterion Layer (B)	Indicator Layer (C)	Method of Calculation	Indicator Meaning	Information Gathering
Space aesthetic value index (A1)	Horizontal space(B1)	Species diversity (C1) *	$H\;=\;-\sum_{i\;=\;1}^{s}{(p}_{i}{\mathrm{lnp}}_{i})$, H is the Shannon–Wiener diversity index, pi is fraction of the entire population made up of species i, S is number of species in the study area.	The Shannon-Wiener diversity index is an index used to investigate the diversity (α-diversity) within the territory of plant communities [59].	Land-use data
		Landscape evenness index (C2) *	$J\;=\;\frac{H}{\mathrm{lnS}\left(\begin{array}{cc} \mathrm{or} & H \end{array}\max\right)}$, J is Pielou’s index of evenness. H is the Shannon–Weiner diversity index and lnS is the natural logarithm of species richness.	Landscape evenness index describes the distribution uniformity of each component in the landscape. The greater the value, the more uniform the distribution of each component in the landscape [59].	Land-use data
		Landscape disturbance index (C3)	HI = $\sum_{i\;=\;1}^{h}\frac{S_{i}}{S_{A}}\times h$, HI is the hemeroby index, h is number of degrees of hemeroby, S_A_ is total area of grid unit, S_i_ is area of cover types with interference level i.	GIS was used to calculate the landscape disturbance index. The hemeroby index (HI) represents the degree of human interference; the smaller the interference intensity, the more conducive to the survival of organisms, the greater the ecological significance of the landscape [60,61].	Land-use data
		Landscape fragmentation index (C4)	C_i_ = N_i_/A_i_, N_i_ is the total number of patches, and A_i_ is the total area.	The fragmentation degree of landscape segmentation reflects the complexity of landscape spatial structure [62].	Land-use data
	Vertical Space (B2)	Vertical visual level (C5)	PA = T/F, T is the sum of diagonal elements, F is the sum of all elements of the matrix.	Vertical visual level represents the degree of superposition of multiple types of landscapes. The stronger the level, the more diverse types, landscape also has a deeper degree of superposition [63].	Panoramic photos
		Color Diversity (C6)	C = σ_rgyb_ + 0.3 × μ_rgyb_, R is red, G is green, B is blue, r_g_ = R − G, r_g_ is the difference between red channel and green channel. y_b_ = 1/2(R + G) − B, y_b_ represents the half of the sum of red and green channels minus the blue channel, $\sigma_{\mathrm{rgyb}}\;=\;\sqrt{\sigma^{2}{}_{\mathrm{rg}}+\sigma^{2}{}_{\mathrm{yb}}}$, $\mu_{\mathrm{rgyb}}\;=\;\sqrt{\mu^{2}{}_{\mathrm{rg}}+\mu^{2}{}_{\mathrm{yb}}}$	Color diversity represents the number of plant colors, such as leaf color [64].	Panoramic photos
		Shape (C7)	${P\;=\;H}_{g}-\frac{\sum H_{i}}{n}$, H_g_ is elevation of center grid, H_i_ is the elevation of the i-th grid in the neighborhood, and n is the number of neighborhood grids.	Shape is based on DEM data and is characterized by calculating the quantitative relationship between the center grid and the neighborhood grid elevation [65].	Elevation data
		Site section geometry (C8)	$\mathrm{Kp}\;=\;-\frac{{\mathrm{rp}}^{2}{+2\mathrm{spq}+\mathrm{tq}}^{2}}{{(p}^{2}{+q}^{2}{)(p}^{2}{+q}^{2}{+1)}^{3/2}}$, p = ∂z/∂x, p is the elevation change rate in x direction, q = ∂z/∂y, q is the y direction elevation change rate, s = ∂2z/∂x∂y, s is the rate of change of height in x direction in y direction, t = ∂2z/∂y2, t is the change rate of height change rate in y direction, r = ∂2z/∂x2, r is the rate of change of elevation in x direction, z = f(x,y), z is the terrain surface function.	Site section geometry is used to describe the curvature variation of a surface curve or surface in the straight direction [66].	Panoramic photos
	Three-dimensional space (B3)	Diffuse non-interceptance (DIFN) (C9)	SVI_n_ = S_n_/A_n_, S_n_ is the number of pixels in the n-th image, and A_n_ is the number of pixels in the sky range in the n-th image.	Diffuse non-interceptance (DIFN) refers to the degree of blue sky that can be seen. It is determined by construction and greening. Open sky visibility can relieve people’s life pressure [67].	Elevation data
		Tridimensional green biomass (C10)	$V\;=\;\sum_{s,t\in D}V_{c}\left(s,t\right)$, $V_{c}\left(s,t\right)\;=\;\sum_{i\;=\;1}^{s}\sum_{j\;=\;1}^{t}\Delta d^{2}\left|h_{ij}-H_{ij}\right|$, V_c_ (s, t) is the tridimensional green biomass of the vegetation in each park, s and t are the number of rows and columns of the image; ∆d is the resolution of the image (0.1 m × 0.1 m); h_ij_ is the actual ground elevation, H_ij_ is the digital surface elevation; D is the spatial extent of each park.	Tridimensional green biomass refers to the space volume occupied by the stems and leaves of all growing plants. According to the high resolution characteristics of satellite images, the tridimensional green biomass of the park is estimated by multiplying the base area by the height [68].	Elevation data
		Canopy density (C11)	A = A_canopy_/A_plot_, A_canopy_ is the projected area of canopy (metres^2^), and A_plot_ is the standard area (metres^2^).	Canopy density refers to the degree to which the crowns of trees in a forest meet each other and shade the ground [69].	Ichnography
		Relief amplitude (C12)	RA = E_max_ − E_min_, E_max_ and E_min_ are the maximum elevation value and the minimum elevation value in each park area.	Relief amplitude is a macroscopic indicator of the topographical features of a region [70].	Elevation data

* The plant species information for this indicator: Deciduous broadleaved forest (arbor), Broad-leaved evergreen forest (arbor), Deciduous needle-leaf forest (arbor), Evergreen coniferous forest (arbor), Deciduous broadleaved forest (shrub), Broad-leaved evergreen forest (shrub), Deciduous needle-leaf forest (shrub), Evergreen coniferous forest (shrub), Grass land.

**Table 3 ijerph-20-03115-t003:** Waterfront park space aesthetic value index calculation results.

Park Name	C1	C2	C3	C4	C5	C6	C7	C8	C9	C10	C11	C12	Average Value
QEP	0.8334	0.8863	0.4344	0.2235	0.2639	0.5455	0.0033	0.8061	0.2237	0.7318	0.7624	0.8536	0.5473
QCP	0.7419	0.8454	0.2459	0.1823	0.4226	0.6363	0.0185	0.8488	0.2648	0.8711	0.5606	0.8394	0.5398
YP	0.7304	0.7863	0.3584	0.1790	0.4545	0.4226	0.0739	0.8032	0.3048	0.6742	0.6126	0.8024	0.5169
WRP	0.6497	0.4525	0.5687	0.1844	0.7142	0.1188	0.0245	0.8085	0.2941	0.8331	0.7043	0.8258	0.5149
YWP	0.7216	0.6012	0.414	0.0708	0.5000	0.3330	0.0080	0.8250	0.3226	0.6942	0.7714	0.8409	0.5086
BTP	0.6817	0.3113	0.5361	0.2357	0.2500	0.6234	0.0172	0.8562	0.3514	0.6804	0.5140	0.8520	0.4925
NWMC	0.6812	0.7063	0.4544	0.1807	0.5454	0.3928	0.0031	0.9351	0.3831	0.1743	0.4820	0.9332	0.4893
CP	0.5214	0.4067	0.3344	0.0660	0.3636	0.4824	0.0206	0.8254	0.2935	0.9015	0.7604	0.8394	0.4846
BP	0.5268	0.3349	0.5319	0.1640	0.5714	0.3376	0.0622	0.8468	0.3132	0.6221	0.6419	0.8373	0.4825
LCP	0.3855	0.4874	0.58369	0.2845	0.7000	0.1911	0.0813	0.8100	0.1426	0.4438	0.8309	0.8095	0.4792
WP	0.1942	0.0654	0.469	0.7467	0.3000	0.6270	0.0648	0.8622	0.7023	0.5627	0.1903	0.8513	0.4697
UBP	0.0533	0.0656	0.4399	0.7677	0.7142	0.4049	−0.0070	0.8564	0.5703	0.5307	0.2732	0.8737	0.4619


 First level park 

 Second level park 

 Third level park 

 Fourth-level park.

**Table 4 ijerph-20-03115-t004:** Horizontal–vertical–three-dimensional spatial index calculation mean.

Park Name	Horizontal Space	Vertical Space	Three-Dimensional Space
QEP	0.5944	0.4047	0.6429
QCP	0.5039	0.4816	0.6340
YP	0.5135	0.4385	0.5985
WRP	0.4638	0.4165	0.6643
YWP	0.4519	0.4165	0.6573
BTP	0.4412	0.4367	0.5995
NWMC	0.5057	0.4691	0.4932
CP	0.3321	0.4230	0.6987
BP	0.3894	0.4545	0.6036
LCP	0.4353	0.4456	0.5567
WP	0.3688	0.4635	0.5767
UBP	0.3316	0.4921	0.5620


 First level park 

 Second level park 

 Third level park 

 Fourth-level park.

**Table 5 ijerph-20-03115-t005:** Landscape visitor evaluation and emotional response to waterfront parks.

Park Name	Landscape Evaluation Number	Park Evaluation Number	Landscape Relative Evaluation Rate	Landscape Emotional Value	Park Emotional Value	Relative Emotional Value
QCP	410	1376	29.8	7.2	3.5	2.1
QEP	26	84	30.1	2.3	3.3	0.7
YWP	10	28	35.7	7.8	4.5	1.7
NWMC	1	3	33.3	3.0	2.0	1.5
YP	7	28	25.0	2.4	1.4	1.7
CP	32	168	19.0	2.5	1.7	1.5
WP	3	30	23.3	0.6	1.5	0.4
BTP	225	630	31.0	5.9	2.1	2.8
BP	27	182	14.8	4.9	3.5	1.4
LCP	22	98	22.4	3.7	3.1	1.2
WRP	2	5	40.0	1.5	0.8	1.9
UBP	185	1822	10.2	2.4	2.8	0.9

**Table 6 ijerph-20-03115-t006:** User sensory evaluation of the park landscape.

Park Name	Vision	Smell	Hearing	Touch	Taste	Feeling
QCP	Wonderful, bright, beautiful scenery	Fresh air, sweet osmanthus fragrance	Good listening, quiet	Smooth	Delicious	Happy
YWP	Beautiful, spectacular	Refreshing, bursts of osmanthus fragrance	Quiet, peaceful, loneness			Beautiful, comfortable, natural and comfortable,
NWMC	Very big	Moldy odor				Not good
YP	Greenery, colorful					Comfort, enjoyable,
CP	Huge, beautiful, golden flower field, green grass	Fresh, urban green aroma	Quiet, peaceful	Soft		Calm, beautiful environment, impressive
BTP	Beautiful, white reed flowers, golden apricot forest, cherry blossom colorful, warm color magnolia	Refreshed	Clean, spring rain drizzling	Swaying in the wind	Tasty	Beautiful, worthwhile trip, poetic, pleasant, still unfinished
BP	Charming, spectacular, endless, colorful falling cherry blossoms, shady trees		Serenity, tranquility	The river wind blows		Safe, comfortable
LCP	National color beautiful, flowers such as a brocade	Day fragrance	Quiet, silent			Good, happy
WRP	Beautiful, good-looking	Fragrant	Tranquility			Leisurely

**Table 7 ijerph-20-03115-t007:** Park thermal value data.

Park Name	Thermal Mean	Thermal Grade
CP	7.1583	High-heat grade
UBP	6.9481	High-heat grade
QEP	5.8863	High-heat grade
QCP	3.3249	Sub-heat grade
BTP	2.1420	Sub-heat grade
BP	2.1073	Sub-heat grade
YP	1.9804	low-heat grade
LCP	1.9525	low-heat grade
WRP	1.6562	low-heat grade
WP	1.6217	low-heat grade
YWP	1.4416	low-heat grade
NWMC	1.3719	low-heat grade

**Table 8 ijerph-20-03115-t008:** User behavior observation record form.

Park Name	Recreation Time (h)	Recreation Number (Persons/Day)	Recreation Frequency (Times/Week)	Recreation Purpose
QCP	1.5	2128	4	Eating, square dancing, passage, fairs, walking
QEP	2	536	4	Viewing, running, basketball, badminton, tennis, walking
YWP	1.5	336	3	Viewing, chatting, walking the dog, picnicking, playing
NWMC	1	384	1	Chatting, accompanying the family
YP	1.5	384	2	Running, cycling, chatting, playing table tennis, walking
CP	3	2160	3	Sightseeing, photography, playing, cycling, running, chatting, playing cards, walking
WP	0.5	680	1	Rope skipping, dog walking, square dancing, Taiji
BTP	2.5	1008	2	Sightseeing, photography, playing, spending time with family
BP	1.5	728	2	Running, cycling, chatting, walking the dog, rope skipping, walking, daze
LCP	1.5	784	1	Viewing, photography, mountaineering, accompanying family
WRP	1.5	552	2	Photography, play, running, square dancing
UBP	0.5	2208	2	Sightseeing, playing, square dance, Taiji, chatting, walking

**Table 9 ijerph-20-03115-t009:** Spatial–psychological–behavioral dimension coupling coordination degree.

Park Name	Coupling Index	Coordination Index	Coupling Coordination Index	Coordination Level	Coupling Coordination Degree
QCP	0.995	0.791	0.887	9	Good coordination
WRP	0.998	0.595	0.770	8	Moderate coordination
BTP	0.898	0.587	0.726	8	Moderate coordination
CP	0.867	0.573	0.705	8	Moderate coordination
QEP	0.746	0.544	0.637	7	Mild coordination
YP	0.946	0.489	0.680	7	Mild coordination
LCP	0.865	0.435	0.614	7	Mild coordination
YWP	0.677	0.382	0.508	6	Poor coordination
BP	0.901	0.266	0.490	5	Near maladjustment
NWMC	0.432	0.265	0.338	4	Mild maladjustment
UBP	0.405	0.263	0.326	4	Mild maladjustment
WP	0.628	0.082	0.226	3	Moderate maladjustment

**Table 10 ijerph-20-03115-t010:** Spatial–psychological dimension coupling coordination degree.

Park Name	Coupling Index	Coordination Index	Coupling Coordination Index	Coordination Level	Coupling Coordination Degree
QCP	0.992	0.804	0.893	9	Good coordination
YP	0.996	0.591	0.767	8	Mild coordination
WRP	1	0.62	0.788	8	Moderate coordination
YWP	1	0.543	0.737	8	Mild coordination
BTP	0.885	0.676	0.773	8	Moderate coordination
QEP	0.645	0.561	0.602	7	Mild coordination
NWMC	0.985	0.392	0.621	7	Severe maladjustment
CP	0.966	0.365	0.594	6	Good coordination
BP	0.966	0.332	0.567	6	Mild coordination
LCP	0.972	0.273	0.515	6	Poor coordination
UBP	0.413	0.112	0.215	3	Moderate coordination
WP	0.576	0.055	0.178	2	Moderate maladjustment

**Table 11 ijerph-20-03115-t011:** Psychological–behavioral dimension coupling coordination degree.

Park Name	Coupling Index	Coordination Index	Coupling Coordination Index	Coordination Level	Coupling Coordination Degree
QCP	0.999	0.735	0.857	9	Good coordination
CP	0.93	0.725	0.821	9	Near maladjustment
WRP	0.998	0.583	0.763	8	Moderate coordination
BTP	0.91	0.7	0.798	8	Moderate maladjustment
LCP	0.922	0.549	0.711	8	Mild coordination
YP	0.951	0.414	0.627	7	Good coordination
QEP	0.809	0.321	0.51	6	Severe maladjustment
UBP	0.893	0.389	0.59	6	Good coordination
YWP	0.593	0.3	0.421	5	Poor coordination
BP	0.857	0.276	0.486	5	Near maladjustment
NWMC	0.289	0.235	0.26	3	Poor coordination
WP	0.506	0.073	0.192	2	Poor coordination

**Table 12 ijerph-20-03115-t012:** Spatial–behavioral dimension coupling coordination degree.

Park Name	Coupling Index	Coordination Index	Coupling Coordination Index	Coordination Level	Coupling Coordination Degree
QCP	0.997	0.834	0.912	10	Good coordination
QEP	0.947	0.75	0.843	9	Moderate coordination
WRP	0.998	0.581	0.761	8	Moderate coordination
CP	0.821	0.63	0.719	8	Moderate maladjustment
YP	0.924	0.464	0.655	7	Mild coordination
BTP	0.998	0.386	0.62	7	Good coordination
LCP	0.822	0.485	0.631	7	Mild maladjustment
YWP	0.591	0.302	0.422	5	Mild coordination
BP	0.955	0.19	0.426	5	Poor coordination
WP	0.988	0.117	0.341	4	Near maladjustment
NWMC	0.341	0.167	0.239	3	Poor coordination
UBP	0.262	0.287	0.274	3	Moderate maladjustment

## Data Availability

The original data are included in the article; further inquiries can be directed to the corresponding author.

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
