# Peer review of "Landscape Aesthetic Value of Waterfront Green Space Based on Space–Psychology–Behavior Dimension: A Case Study along Qiantang River (Hangzhou Section)"

_ijerph, 2023, doi:10.3390/ijerph20043115_

Round 1

Reviewer 1 Report

This article presents the idea of a multi-faceted assessment of the aesthetic value of green areas along a riverbank in a city. This has practical significance for people living in the city, as well as for tourists visiting the city. In turn, the results can inspire the managers of these parks in their work to improve their management. However, the work has many shortcomings that prevent it from being published. The methodology is very superficially described, sometimes it is difficult to see where the figures presented in the results come from. In addition, there are errors in the results, which calls into question the value of all the data presented. This impression is compounded by the shortcomings of incorrectly numbered tables and figures and the lack of titles in some of them. Inference from the results presented is also flawed in many cases. Some sentences are very long, making them uncomfortable to read. I provide detailed comments on the text below.

Title

It is quite long and complicated, perhaps it could be shortened?

Authors

Is the addition of the "&" necessary here? After all, there is an "Author Contributions" paragraph at the end of the article, that's where this information should be

Abstract

L10 - redundant second word "urban"

Keywords

There are a lot of them. It seems to me that the following can be omitted as being too general: text sentiment analysis; word frequency analysis; kernel density analysis

Introduction

L38 - unnecessary second word 'urban'

L39-42 - and what about enhancement and protection of biodiversity?

L45-46 - please write more about have these positive impacts (space for jogging, walking? etc). I think you can also use knowledge about ecosystem services here (MEA 2005 - https://www.millenniumassessment.org/en/index.html)

L71-79 - these are not sentences, verbs are missing

L90-97 - this is more of a description of materials and methods. Here it would be appropriate to state the research problem the authors want to solve, the objectives to be achieved both theoretically and practically. Why and for what purpose was this research undertaken by the authors?

Materials and Methods

There are a lot of errors in the use of spaces.

L101 - is Hangzhou the name of the city? because it is not clear

L116-118 - problem with the construction of this part of the sentence

Fig. 1 - too small, completely illegible captions. Please move the fourth image under the first three and enlarge them all significantly to the width of the page - there is a chance they will then be readable. Is it not possible to mark the boundaries of the sites on the fourth image? One number on a map when the site in question is 1.3 million m2 is not very convincing.

Table 1 - I would suggest putting "Hangzhou City, Zhejiang Province" in the title of the table so that it is not repeated in all the rows of the table. Instead, it would be worth adding the geographical coordinates of the centre of these objects in the Location column, as stating that they are on a street is not very specific, especially if the street is long. Area - wouldn't it be simpler to give in ha?

Fig. 2 - here it is City Balcony Park, and earlier it mentions Urban BP

Fig. 3 - perhaps it would be worth rotating 90 degrees and enlarging to the full page to make it easier to read? First module - it may be worth separating the indicators for each criterion more spatially. Title of this module - it may be worth using similar terms as in Table 2 (criterion/indicator). Second module - what do the dots at the bottom of Crowd experience mean? if Others, it should be written as in the third module. The arrowheads at the bottom of the figure are not visible. Further in the Materials and Methods section there is no explanation for the lowest part of Figure 3

L136 - maybe: OF Landscape...? (same further on in Results chapter)

Table 2 - citations for the various solutions adopted are missing. There should also be a copy of the table heading on each subsequent page.

A1 - this is quite complex wording - maybe it can be simplified?

C1 - what species were considered - all possible (plants, animals, fungi) or plants alone? in Margalef index is lnN, not lnA (i.e. refers to number of individuals, not area of site). Why is the number of species in the community (which community?) and not the number of species in the study area? where will this data be taken from? there is no information on this in the article

C2 - it is not clear what the observed species diversity index is and what the maximum species diversity index is and how this differs from C1. What species are taken into account? Where will this data be taken from? there is no information on this in the article

C3 - what corridor is involved? a space devoid of a given ecosystem? then why is it length and not area? where will this data be taken from? there is no information on this in the article

C7 - method of calculation - shouldn't the sign of the sum be just before Hi? additionally: probably the word "slope" is redundant

C8 - method of calculation - the symbols "z" and "h" are missing in the explanations

C9 - method of calculation - the inclusion of the symbols "Sn" and "An" in the explanations is missing. "the degree of blue sky that can be seen" strongly depends on how the photo is taken, for example....

C10 - method of calculation - the inclusion of the symbol "Vc" in the explanations is missing, the green biomass is also poorly indicated. which is supposed to be the result of this formula

C12 - what is meant by unit area? the whole park analysed?

L147 - maybe: OF Landscape...? (same further in Results chapter)

L149 - only tourists? and what about local peoples, who spend their leisure time here? maybe a better term would be "users"?

L151 - which blog?

L152-157 - repeated sentences and fragments of sentences

The description of the Psychological dimension of landscape aesthetics research lacks reference to the right part of this module in Fig. 3 (crowd experience)

L176 - maybe: OF Landscape...? (same further on in chapter Results)

Subchapter 2.3 - here should be references to all indicators C1-C12 in turn - how the information for their calculation was collected

L197 - the word average is redundant and even wrong

Panoramic photos - was a photo taken in each 15x15 or 30x30 field? in which part of it?

Weibo comments - how was this information retrieved, what period was it from (if it's from a number of years ago, these parks may have looked a bit different), was exactly all found taken into account?

Crowd thermal data - why wasn't this research done within the same period as the Crowd recreation behaviour research? these could be different groups of park users. In Europe, July and August are the holiday months and you are more likely to see tourists in cities whose purpose for visiting riverside areas may be different to people (city dwellers) who will be back in town from the start of September

L213-214 - this is not a sentence. How was recreation time determined? were people using the park tracked or were they interviewed at the entrance as to how much time they planned to spend (and how)?

Results

L221-225 - this is a methodology rather than a result. Furthermore, there is too little detail given as to how this was technically carried out, what criteria/problems/assumptions were taken into account when interpreting the images, it is completely unclear where these values in Table "4" come from (nota bene this is after all Table 3, not 4)

L228 - why such a range of values? not at all apparent in Figure 5 (there is an error in the figure number in L228). "Among them" suggests that we continue to talk about this range, meanwhile the quoted QEP park is already outside this range - this is inconsistent. Additionally, there is an error - in the graph this highest value is QCP. Please check everywhere for correct data and calculations for QEP and QCP.

L231 - 41.6% of what? number of parks? their area?

Table "4" - it is possible to set these pairs in the same order as in Table 1, as there is some different order each time (or in all tables set the pairs according to the numbering in Fig 1). Alternatively, another solution could be used: set the rows in Table "4" according to decreasing Average Value, this would immediately replace Fig. 5 (you could then mark the rows in the table with these different colours as in Fig 5, and in the footnotes explain these colours)

L241 - redundant "The results show that:"

L244 - 0.3666 not 0.3880

L245-247 - % of what?

L248 - error either in the description or in Table "5" (which should be #4) - CP has a value of 0.3666 in the Table and here 0.3880

L249 - error either in the description or in Table "5" - the highest value is 0.4691, not 0.4630. Imprecise term "the highest park" (as well as the lowest)

L256 - this is where Subchapter 3.2 should Begin

L256-259 - this is a methodology, not a result, and not very detailed anyway ("that did not meet the original requirements and were meaningless" - no information on these details, assumptions)

L261 - "sentiment values" - there is no such category in Table 6?1?5?

L262 - "Attractions evaluation of emotional value analysis results" - not a very understandable statement

L263 - simplification (the highest park, the lowest park)

L263-265 and Table 6?1?5? - statements are used here and calculations are given which are based on only a very few opinions (1, 2, 3, 7) - this is not a particularly comprehensive source of information, and nowhere in the text did I find a disclaimer that perhaps the results would have been different if there had been more comments examined

L268 - "relative landscape value" - this term is not in Table 6?1?5?

L277 - should be Figure 6 (here and in its description)

Fig 5?6? - equation describing the line is missing

L284 - should be Table 6

L284-286 - "The results show that the park with emotional value greater than 2 ( Qianjiang Century Park, Baita Park ) has a rich and comprehensive sensory experience" - this is rather not because the score is >2, but because there are the most comments for them and therefore the highest probability of finding terms relating to all senses and feelings. Similarly, a low number of comments means that not all senses will surface, from which one cannot conclude that the park is worse value in this respect. This is an erroneous inference

L288 - should be between 1-2, not 0-1

Table 7?6? - LCP - day fragrance should be in the next column rather

L295-300 - this is the methodology. In addition, the table numbering error continues (I am not mentioning this error any further, but it still occurs)

L298-299 - why is the character ~ used ? here it should be precise, not "about 5"

L306-307 - "Sub-heat grade includes QEP, WP, BP and QCP." - the authors forgot BTP

L308-309 - not a very understandable sentence. Also, the description under the table pretty much repeats what is in the table, so the question is whether it is that necessary

L313-316 and Table '9' - it is unclear how the information was obtained that a person e.g. 4 times/week is in the park, that he or she spends 1.5 hours in the park etc. The fact that someone eats is of course evident, but the frequency of stay and time spent in the park are not such obvious data anymore (without tracking each person, and this was unlikely to be done...). Hence my question - how was this measured?

L320 and Fig "1" - check the numbering of Figures. What is the unit of this density?

L323 - "among them" suggests parks in the range 0.0401 to 0.0177, and the parks given next are also outside this range

Fig "1" - no correct description

In the last Subchapter of results there is no reference to differences in results by time of day and day of week - the methodology indicated that surveys were conducted taking into account their variation

Discussion

The main shortcoming of the discussion is the lack of any reference to the literature. In fact, it has more the appearance of a further sequence of research findings.

L334-335 - passage hard to understand (the Qiantang River Hangzhou section waterfront park space-psychology-behaviour dimension overall coupling coordination degree distribution)

Tables "10-13" - shouldn't "coordination index" be there instead of "coordinated index"?

L363-365 - note that some of the conclusions were drawn from a small number of statements

L368 - there is an error here or in Table 13, QEP has a high score here and in Table there is only 5. For this, the higher score is BTP, which is missing in L368-369

L377 - 'low north bank, high south bank' - I would put this more carefully as there is not some extreme difference (e.g. lower north bank, higher south bank)

L381-382 - this is not shown on any summaries/tables

L382-384 - "From the perspective of urban development, waterfront parks close to the city 's economic centre (such as QCP, CP) are highly coupled and coordinated with waterfront parks in other areas". - I don't really understand this passage, after all wasn't there an analysis of the relationship between parks? "Close to the city's economic centre" - it is worth indicating the boundaries of this centre in Fig. '7'

Fig. "7" - it is worth enlarging, because in the usual 100% dimension not much can be seen

Conclusions

The first paragraph will be justified as the description of the methodology will be improved, the results will be fairly shown and there will be a wider discussion of the results.

1 - there was no information in the text that the average evaluation index is 0.5168. The highest QEP or QCP? L413-416 - this is not a sentence, in addition it is vaguely and heavily worded

2 - L420 - 25% is probably not "proportion of landscape emotional value greater than 5" but proportion of parks with that value... Furthermore, is there not already too much wording for the same thing at once here: "landscape emotional value greater than 5 with positive emotional value"? L422-424 - as I pointed out earlier, this is also influenced by the number of comments

3 - L432 - there is no such thing in Table 1 and throughout the text as "China Water Expo"

4 - "high north bank and low south bank" - a simplification, plus the differences are not that dramatic. "high city and low suburb" - this was poorly shown in Fig "7" - after all, there are parks at the ends of the range with a value of 8, so quite good. And the ends of the study area range are probably the suburbs. L444 - error - instead of QEP it should be BTP

Last paragraph - partly repeats information from first paragraph. In addition it will be true if you remove all the shortcomings from the text and especially give the methodology more accurately. Anyway, there are also "the shortcomings of less data points" here in the sense of too few comments for some parks

Conflict of interest

typos

Abbreviations

DIFN and DEM have the same formulation

References

There are few references from outside China - is it really so difficult to find articles on this subject from other regions? There are also many missing capital letters in proper names

8 and 47 - there seems to be a problem with names

59 - redundant one 59

Reviewer 2 Report

The topic of the manuscript is interesting, and it is based on a large data set, with relatively clear research purpose. However, the maunuscript is poorly writen, with many mistakes and improper statement. The methods used for data analysis are quite simple thus more potential of the research was not explored. Several tables and figures are not efficiently presented, etc......The authors need to put much more efforts on the manuscript before it could be published. Some suggestions are listed below.

(1) The title could be Landscape Aesthetic Value of Waterfront Green Space based on Space-Psychology-Behavior Dimension: A Case Study along Qiantang River (Hangzhou Section)”.

(2) More detailed and precise information about the results should be provided in the abstract.

(3) Line 47-49, it is not an objective and reasonable statement.

What is difference between ”waterfront green space” and blue space? How to exactly classify them in a landscape with both green space and blue space? The author should explain why the study is important from this perspective in the introduction.

(4) Clear scientific questions should be proposed in the introduction.

(5) The main figure in Fig.1 is too small.

(6) Name of Table 1 is improper, and the location information is not necessary, and many other landscape information about the parks is missing.

(7) Spatial dimension in Fig.3 is not clear. The indicators corresponding to the three types space should be in consistent with Table 2.

(8) Line 137-138, do you have any references?

(9) Line 197-198, I am not convinced by the method to deal with the data with different standard in a study.

(10) The information in Fig.5 and Table 3 are duplicated and not in consistent (QCP, QEP).  

(11) The language of this manuscript has big problems and needs substential professional improvement, such as:

1) What do you mean by “uneven distribution of aesthetic value” and “uneven landscape effect” ? It is quite hard to understand in English.

2) Line 13-20, this is a very long sentence with improper grammar use.

3) Line 83-84, the future cannot build...

4) Line 289, etc..

Reviewer 3 Report

Thank you for giving me this opportunity to read the manuscript entitled "Study on Aesthetic Value of Waterfront Green Space Landscape Based on Space-Psychology-Behavior Dimension: A Case Study of Qiantang River (Hangzhou Section)". The topic of this manuscript is interesting and would be a good contribution to this field. I think it could be considered for publication in IJERPH once the following issues are addressed.

1.     Please replace the keywords with too many words, as short keywords will facilitate your paper being searched by potential readers.

2.     Texts in Figure 1 should be enlarged to make sure they can be clearly read.

3.     Lines 51-52 “, which leads to the uneven distribution of social public resources [6-9],”: a newly published paper titled “Observed inequality in urban greenspace exposure in China.” is suggested to be cited as a reference.

4.     Limitation section should be added as a sub-section to the Discussion.

5.      Some grammatical errors exist in the manuscript. Therefore, a critical review of the manuscript's language will improve its readability.

Round 2

Reviewer 1 Report

The authors have done a tremendous and patient job of improving the article according to the comments I made in my first review. My biggest concern (and objection to the publication of this article), however, is that the Methodology has partly provided different formulae, as well as the dates on which some of the analyses were carried out (thus a different set of analysed data), while the results have remained identical to the first version. I provide detailed comments on the text below.

Abstract

L9-11 - I would add the word "sometimes" in this sentence, as not every "green space with an excellent landscape aesthetic value fails to serve the needs of most citizens"

L22 - unexplained abbreviations of QEP and UBP. Furthermore, this is an oversimplification: "QEP was the highest" - rather "QEP obtained the highest value" or something like that

Introduction

L56 – which city?

L55-61 – the following sentences are not compatible with each other: „Because of the present parallel layout of blue space and green space in the coastal city, the landscape effect of urban waterfront green space is unbalanced and monotonous” vs „However, at this stage, the landscape effect of urban waterfront green space varies”. In the second sentence I would add "sometimes"

Materials and Methods

Fig. 1 - the park numbers on the bottom map are invisible - please very much enlarge them and change the font colour to white. It is also worth reinforcing park boundaries as they are not visible

Table 1 - under the heading GEOGRAPHICAL COORDINATES please add in brackets what the geographical directions are (e.g. N, E)

L146 - "Homogeneous heterogeneity" - are you sure this term was meant to be here? And not the heterogeneous isomorphism of Fig. 3?

Table 2 - there should be a copied table heading on each subsequent page. The authors have changed some of the calculation formulae used and the results (Table 3) remain the same - this is rather unrealistic

C1 - method of calculation - Wiener not Weiner; indicator meaning - is it definitely supposed to be Simpson's index here? Still missing from the description is the information that only plants were considered. Did the data for this formula really come from land-use data (diversity and number of individuals of plant species)? this would rather be visible on the orthophotomaps

C2 - please add information that plants were taken into account

L206 and 208 - I understand you mean 'm' (metres)?

L232 - the authors changed the period of the crowd thermal date survey by a month and the results are identical - after all, this is unrealistic

Results

L243-250 - this should be in Materials and Methods

L252 - I have a doubt about the first number - where did it come from? there will probably be less doubt if it is worded as follows: "were concentrated in the 0.4649-0.5744 range".

Tables 3 and 4 - footnote - maybe instead of secondary park give second level park, it will be compatible with the other descriptions

L287 - instead of sentiment it should be emotional

L288 - "the highest was YWP, the lowest was WRP". - mental abbreviation, it is more about the value for these parks

L317-320 - this is rather a description of the methodology, it should be in another chapter

L339 - should be 0.0011-0.0671 persons/m2

Fig. 6 - should read (persons/m2) in the caption

Discussion

Fig. 7 and Table 9 - there is a complete disagreement between the values for the individual parks shown above the bars with those written under the map and those in the table!!! Which of these are correct?

L423-432 - also missing here is the statement that some of the calculations are based on very few opinions about the parks (Table 5), this too should be verified in the future as the results obtained so far may not be very representative

Conclusions

L434-447 - should be 'dimension of...', as in the titles of earlier subsections

L459-462 - it should be added here that this also has a connection to the large number of comments available about the park

Reviewer 2 Report

The authors have addressed most of my concerns. It could be suitable for publication.

Author Response

Thank you for your advice and affirmation.